# Helminth Coinfections Modulate Disease Dynamics and Vaccination Success in the Era of Emerging Infectious Diseases

**DOI:** 10.3390/vaccines13050436

**Published:** 2025-04-22

**Authors:** Brice Armel Nembot Fogang, Linda Batsa Debrah, Michael Owusu, George Agyei, Julia Meyer, Jonathan Mawutor Gmanyami, Manuel Ritter, Kathrin Arndts, Derrick Adu Mensah, Tomabu Adjobimey, Achim Hörauf, Alexander Yaw Debrah

**Affiliations:** 1Department of Clinical Microbiology, School of Medical Sciences, Kwame Nkrumah University of Science and Technology, Kumasi 00233, Ghana; lbdebrah.chs@knust.edu.gh (L.B.D.); gagyei7@st.knust.edu.gh (G.A.); derrickadumensah@yahoo.com (D.A.M.); 2German-West African Centre for Global Health and Pandemic Prevention (G-WAC), Partner Site Kumasi, Kumasi 03220, Ghana; mowusu55@knust.edu.gh (M.O.); jgmanyami@gmail.com (J.M.G.); 3Institute for Medical Microbiology, Immunology and Parasitology (IMMIP), University Hospital Bonn (UKB), 53127 Bonn, Germany; julia.meyer@ukbonn.de (J.M.); manuel.ritter@ukbonn.de (M.R.); kathrin.arndts@ukbonn.de (K.A.); tomabuadjobi@hotmail.com (T.A.); achim.hoerauf@ukbonn.de (A.H.); 4Kumasi Centre for Collaborative Research in Tropical Medicine, Kwame Nkrumah University of Science and Technology, Kumasi 00233, Ghana; 5Department of Medical Diagnostics, Kwame Nkrumah University of Science and Technology, Kumasi 00233, Ghana; 6School of Public Health, Kwame Nkrumah University of Science and Technology, Kumasi 00233, Ghana; 7German-West African Centre for Global Health and Pandemic Prevention (G-WAC), Partner Site Bonn, 53127 Bonn, Germany; 8Department of Medical Laboratory Technology, Royal Ann College of Health, Kumasi 00233, Ghana; 9Department of Public Health Education, Akenten Appiah-Menka University of Skills Training and Entrepreneurial Development, Mampong 00032, Ghana; 10Faculté des Sciences et Techniques (FAST), Université d’Abomey Calavi, Abomey Calavi BP 526, Benin; 11German Centre for Infection Research (DZIF), Neglected Tropical Disease, Partner Site Bonn-Cologne, 53127 Bonn, Germany

**Keywords:** helminth, co-infection, infectious diseases, non-communicable diseases, immune response, vaccine efficacy, outcome

## Abstract

**Background/Objectives**: Helminth infections, particularly prevalent in low- and middle-income countries, have been extensively studied for their effects on human health. With the emergence of new infectious diseases like SARS-CoV-2 and Ebola, their impact on disease outcomes become more apparent. While individual studies have explored the impact of helminth co-infections on disease severity and vaccine efficacy, the findings are often inconsistent and context-dependent. Furthermore, the long-term effects of helminth-mediated immunosuppression on vaccine efficacy and its broader implications for co-infections in endemic regions remain not fully understood. **Methods**: This systematic review conducted in line with the Preferred Reporting Items for Systematic Review and Meta-Analysis (PRISMA) 2020 guidelines synthesizes the current evidence, identifies patterns, and highlights areas needing further research, offering a cohesive understanding of the topic. PubMed, Scopus, Google Scholar, and Cochrane Library were searched to include studies published from 2003 to February 2025. **Results**: Co-infection reveals a dual role of helminths in modulating immune responses, with both beneficial and detrimental interactions reported across studies. It may confer benefits against respiratory viral infections by muting hyper-inflammation associated with the severity of conditions like COVID-19, Influenza, and RSV. However, they can exacerbate disease outcomes in most bacteria and blood-borne viral conditions by impairing immune functions, such as neutrophil recruitment and antibody response, leading to more severe infections and higher viral loads. The stage of helminth infection also appears critical, with early-stage infections sometimes offering protection, while late-stage infections may worsen disease outcomes. Helminth infection can also negatively impact vaccine efficacy by suppressing B cell activity, reducing antibody levels, and decreasing vaccine effectiveness against infectious diseases. This immunosuppressive effect may persist after deworming, complicating efforts to restore vaccine efficacy. Maternal helminth infections also significantly influence neonatal immunity, affecting newborn vaccine responses. **Conclusions**: There is a need for targeted interventions and further research in helminth-endemic regions to mitigate the adverse effects on vaccine efficacy and improve public health outcomes.

## 1. Introduction

Parasitic worms, collectively known as helminths, have shared a complex and enduring relationship with humans that stretches back millennia [1,2]. Fossilized evidence, such as a *Schistosoma mansoni* egg discovered in the remains of a 6200-year-old human skeleton, underscores this long-standing coexistence [3]. These parasites, classified into three major groups—nematodes, trematodes, and cestodes—have co-evolved with their hosts, driving an evolutionary arms race that has profoundly shaped both their survival strategies and the human immune response [4]. Today, helminths remain a significant public health concern, particularly in sub-Saharan Africa [5], where they infect an estimated 1.5 billion people and contribute to 85% of the global burden of neglected tropical diseases (NTDs) [6]. By inducing malnutrition, anemia, and chronic illnesses, they exacerbate poverty cycles and strain health systems [7]. Beyond their direct health impacts, they are renowned for their ability to manipulate the host immune system with profound implications on vaccine efficacy and successes in endemic regions. Their skewed immune responses toward anti-inflammatory Th2 and regulatory pathways are known to suppress pro-inflammatory Th1 and Th17 responses [8,9,10,11]. This immunomodulation allows them to establish chronic infections and influences the outcomes of co-infections [12].

Studies examining the impact of helminth infections on viral diseases in endemic areas where coinfections are particularly common have yielded varied results, reflecting the complexity of these interactions [13]. They significantly alter immune responses, potentially benefiting or hindering the host’s ability to combat bacterial and viral pathogens [14,15]. They have been shown to suppress hyperinflammatory responses, mitigating the severity of respiratory virus infections including SARS-CoV-2, Influenza, and RSV (Respiratory Syncytial Virus). We have previously demonstrated that helminths’ seropositivity correlates inversely with Th1 and Th17 cytokines and severe COVID-19 [16]. However, the opposite side has potential disadvantages, as helminth infections were associated with diminished antibody response, reduced vaccine efficacy, increased susceptibility to concurrent infections, and potential interference with tumor immunosurveillance [17,18,19,20,21,22,23,24,25,26,27,28]. While deworming has clear benefits for reducing helminth-associated morbidity, its modulatory effect is resilient and persists in the host even after deworming, posing a new challenge of vaccination timing post-deworming [13]. This dynamic relationship has not only influenced the human immune landscape and susceptibility to infectious diseases but also non-communicable diseases (NCDs), including type 2 diabetes, allergic diseases, and autoimmune infections. Sub-Saharan Africa, where helminth infections are widespread, has reported surprisingly lower incidence and less severe COVID-19 outcomes, raising questions about the role of helminths in modulating immune responses to the virus. Although this might be associated with the lower testing capacity or the younger demography of the continent, recent studies showed a negative association between *Ascaris lumbricoides* seropositivity and COVID-19 severity in endemic countries [29]. Moreover, research has demonstrated that helminth antigens from *Ascaris lumbricoides*, *Onchocerca volvulus*, and *Brugia malayi* differentially modulate the activation of the CD4+ and CD8+ T lymphocytes of convalescent COVID-19 patients in vitro, highlighting their potential to influence viral immunity [30]. Despite these findings, the broader implications of helminth co-infections on viral immunity and vaccine efficacy, including COVID-19, remain poorly understood [31]

Building on this foundation, this present review aims to make a synthesis of recent research on helminth coinfections and their impact on disease outcomes and vaccine efficacy, thereby facilitating a thorough comprehension of how these interactions may affect health outcomes and shape the immune response. This exploration not only sheds light on the intricate dynamics of helminth–host–pathogen interactions but also offers a promising avenue for rethinking approaches to disease prevention and treatment in areas where these parasites are endemic.

## 2. Materials and Methods

A comprehensive literature review was conducted with adherence to the preferred reporting items for systematic review and meta-analysis (PRISMA) guidelines (Appendix A) to summarize the findings on the influence of helminths on communicable and NCD outcomes as well as vaccine efficacy.

### 2.1. Protocol Registration

The protocol for conducting this systematic review was registered and approved at the International Prospective Register of Systematic Reviews, PROSPERO registration number CRD42024556697.

### 2.2. Searches

The search encompassed multiple electronic bibliographic databases and aggregators, including Scopus, PubMed, Google Scholar, and Cochrane Library. Additionally, the reference list of included studies was screened for relevant articles. The search strategy was structured around key concepts essential to the review: helminth co-infection, disease dynamics, vaccination success, and infectious and non-infectious diseases. Each key term was defined by a range of synonyms and adapted for suitability across different databases. Boolean operators such as “AND” and “OR” were used to optimize the search to find articles relevant to the search question. The timeframe from 2003 to February 2025 was selected to capture recent advancements in understanding helminth co-infection. The search terms targeted studies reporting the interaction between helminths and other disease conditions and vaccine efficacy. The search did not impose any language restrictions.

### 2.3. Study Selection Procedures

The inclusion and exclusion criteria were established using the participants, intervention/exposure, comparator, and outcome (PICO) framework, as outlined below.

#### 2.3.1. Participants/Population

This review included studies conducted at the population or cohort level, regardless of the administrative scale (district, regional, or national) on individuals with bacterial infections, viral infections, NCD, or vaccinated individuals.

#### 2.3.2. Intervention(s)/Exposure

The primary exposure of interest was the co-occurrence of helminth infections with viral or bacterial infections or non-communicable diseases (NCDs), or vaccine administration.

#### 2.3.3. Comparator(s)/Control

The comparator group consisted of individuals with mono-exposure (without helminth co-infection) to viral or bacterial infections or non-communicable diseases (NCDs) or vaccines.

#### 2.3.4. Main Outcome

The primary objectives of this review focused on clinical effects (disease severity), immunological effects (cytokine expression, T cell activation, antibody production), and vaccine efficacy in coinfection. A narrative summary of the key findings from the selected articles is provided, highlighting reported disease outcomes and immune parameters.

### 2.4. Eligibility Criteria

Studies reporting on immune responses and/or disease outcomes in individuals with helminth co-infections.Population-level, cohort-based, or facility-based studies conducted at any administrative level (district, regional, or national).Studies including a comparator group assessing immune responses and/or disease outcomes in participants without helminth exposure or infection.

### 2.5. Study Inclusion/Exclusion

Two independent investigators applied eligibility criteria to select studies for inclusion in this review. Disagreements were resolved through discussion, with a third reviewer consulted to reach consensus when necessary. Studies were included if they provided the following: data on immune responses and/or disease outcomes in scenarios involving helminths with viral or bacterial infections, vaccine efficacy, or non-communicable diseases (NCDs) under mono-infection and co-infection conditions; a documented methodology for evaluating immune responses and/or disease severity; a defined study population size; and observed outcomes related to disease severity and/or immune response. Studies were excluded if helminth infection status was not assessed or was only evaluated post-vaccination, or if a comparative control group was not included.

### 2.6. Data Extraction

The following data were extracted: author(s), publication year, immune response, clinical outcome, and study type (human, animal model); type of population (children, adults, pregnant women); and type of coinfection (virus, bacteria, NCD). Mendeley Desktop Version 1.19.8 was used to identify duplicate records.

### 2.7. Measures of Effect

This review’s primary objective was to estimate the immune response and/or disease outcome (severity) as reported in primary studies. This review encompassed both observational and intervention studies.

### 2.8. Article Screening

The review process began with the identification and removal of duplicates. Titles and abstracts of the retrieved articles were then screened to determine eligibility for inclusion in human and animal studies. In the third stage, full texts of articles identified as relevant during the second stage were thoroughly reviewed. At each stage, two reviewers independently assessed the articles, with a third reviewer resolving any disagreements through discussion to reach a consensus. Studies were included if they compared immune responses, disease outcomes, or vaccine efficacy between helminth-infected and uninfected groups, or between groups treated and untreated with anthelmintics. Additionally, the helminth status of participants had to be laboratory-confirmed prior to vaccination, with the subsequent measurement of immunological outcomes.

### 2.9. Data Analysis and Quality Assessment

Immune responses and disease outcomes were organized in a tabular format and analyzed narratively, with the findings synthesized into thematic narratives. Data from relevant articles were extracted from texts, tables, and figures into a custom Microsoft Excel tool. Extracted data included study type and participant characteristics; details on viral infections, bacterial infections, and non-communicable diseases; vaccines; helminth species or anthelmintic treatments; and clinical or immunological outcomes. The quality of the studies was assessed using the Effective Public Health Practice Project (EPHPP) tool, which rated studies as strong, moderate, or weak. This assessment was based on an eight-component checklist: selection bias, study design, confounders, blinding, data collection methods, withdrawals and dropouts, intervention integrity, and analyses. Two independent reviewers used this tool to assess the quality of each study and any disagreement was settled by a third reviewer.

## 3. Results

The eligibility and screening process for the articles included in this review is comprehensively detailed in Figure 1. This figure not only presents the inclusion and exclusion criteria but also provides an explanation of the reasons for excluding specific studies.

A total of 78 published articles were selected for inclusion in the review. Of these, 32/78 of the studies involved animal models, whereas 46/78 studies involved human subjects (Figure 2). Within the subset of human studies, 5/46 investigated the impact of maternal helminth infection on the immune responses of offspring. The remaining 41/46 studies assessed the effects of direct helminth exposure on other infections and vaccine outcomes. A total of 59 of all included 78 studies investigated the effects of helminths on infectious and non-infectious disease outcomes, with these studies being further categorized based on the type of co-infection: 29/59 on viral co-infections, 10/59 on bacterial co-infections, and 20/59 on non-communicable disease co-infections (Figure 2). Furthermore, 19/78 of the included studies examined the influence of helminths on vaccine efficacy. This group consisted of ten (10) human studies and nine (9) animal studies. The studies collectively reported on vaccine efficacy against several diseases, including COVID-19, Ebola, tuberculosis, Influenza, tetanus, typhoid fever, malaria, Human Papillomavirus infection (HPV), yellow fever, measles, diphtheria, and HBV infection (Figure 2).

### 3.1. Helminth and Emerging Respiratory Viruses

Several reviewed articles assessed the interaction between helminths and emerging viral disease infections. Most studies focused on the interaction between helminths and SARS-CoV-2, followed by Respiratory Syncytial Virus (RSV) and Influenza virus. The interaction between helminth infections and respiratory viruses reveals a complex but potentially beneficial relationship in terms of viral disease outcomes (Table 1). Most reviewed studies suggested a potential modulatory interaction between helminths and respiratory viruses, leading to less severe respiratory disease outcomes in individuals who were co-infected [29]. Studies indicate that patients with parasitic helminths and protozoans like *Entamoeba* spp., *Hymenolepis nana*, *Schistosoma mansoni*, *Trichuris trichiura*, and *Ascaris lumbricoides* are less likely to experience severe COVID-19 [29]. Most viral infections, including SARS-CoV-2 and MERS-CoV, have been associated with hyperinflammation and the down-regulation of Th2 cytokines, leading to severe disease outcomes and high fatality rates [32,33]. The modulatory effect of helminths appears to be mediated by helminth antigens, which dampen the immune response by reducing pro-inflammatory response while increasing the anti-inflammatory response. Higher levels of *Ascaris* antibodies were associated with asymptomatic SARS-CoV-2 infection and reduced risk of severe disease, likely due to lower systemic inflammation [29]. In the case of Influenza, most animal model studies seem to converge that *Heligmosomoides polygyrus bakeri* (Hpb) infection in mice leads to better survival and less weight loss, attributed to enhanced lung immune responses, lower viral loads, and reduced inflammation [34]. However, the outcome of the interaction between filarial infections and Influenza is stage-dependent; early infections lead to less severe symptoms, while late-stage infections worsen disease outcomes [35]. Like COVID-19, *H. polygyrus* co-infection with RSV (Respiratory Syncytial Virus) results in less severe disease and lung inflammation in mice, driven by an increase in monocytes in the blood and lungs through type 1 interferon signaling, with severity mitigation requiring normal microbiota but not a Th2 response [36,37]. Reduced severity of respiratory viruses seems to be primarily associated with a skewed Th2 response associated with chronic helminth infection. In addition to the skewed helminth Th2 response, helminth-derived products such as cystatins, omega-1 glycoprotein, and ES-62 are potent immune modulators and can potentially affect host immune responses and viral disease outcomes [38]. These products, largely produced by many helminths, including filarial nematodes, can interfere with NF-κB (Nuclear Factor kappa-light-chain-enhancer of activated B cells) pathway activation, inhibiting proinflammatory cytokines (Figure 3). NF-κB and IRF3 (Interferon Regulatory Factor 3) are key players in the innate immune response, particularly in response to viral infections, where they promote the Th1 response. NF-κB activation is stimulated by proinflammatory cytokines like TNF-α and IL-1, which bind to cell receptors and initiate a signaling cascade, leading to IκB (Inhibitor of Nuclear Factor kappa B) degradation and NF-κB activation [39]. Helminths, however, can suppress or alternatively activate this signaling pathway by secreting immunomodulatory molecules that prevent IκB degradation and block NF-κB nuclear translocation or promote the transcription of Th2 molecules, thereby reducing inflammation. Inhibition of Toll-like receptor (TLR) signaling by helminths can also lead to the same effect [38,39]. Helminth-derived products containing glycoprotein also have the potential to interfere with TLR pathways. ES-62 modulates TLR pathways, particularly TLR4, to disrupt downstream NF-κB activation by interfering with kinases and transcription factors [40]. This molecule secreted by *Fasciola hepatica* inhibits TLR3 and TLR4, affecting TRIF (TIR domain-containing adapter-inducing interferon-β) and MyD88 (Myeloid Differentiation Primary Response 88)-signaling pathways. These strategies reduce NF-κB activation and cytokine production, promoting an anti-inflammatory environment [41,42]. ES-62 can also induce a semi-mature phenotype in dendritic cells (DCs), decreasing the expression of MHC II molecules on their surface. This reduction in MHC II expression promotes the differentiation of regulatory T cells (Tregs), which in turn suppresses Th1 and Th17 cells. Cystatins and glycans, both powerful immune modulators, promote M2 macrophage polarization, which enhances the production of Th2 cytokines that are anti-inflammatory. Glycans can also directly suppress IL-12 production, further supporting an anti-inflammatory response. Additionally, TGF-β mimics and miRNAs act as potent modulators: TGF-β mimics engage TGF-β receptors on Tregs and macrophages, promoting Treg expansion and M2 macrophage polarization, which further drives IL-10 and TGF-β production. These cytokines inhibit neutrophil extracellular traps (NETs), leading to a more anti-inflammatory environment and the suppression of Th1 and Th17 cytokines such as IFN-γ, TNF-α, IL-17, and IL-12. Meanwhile, miRNAs support the proliferation of both Tregs and B regulatory (Breg) cells, inhibit effector T cells (particularly Th1 and Th17), and modulate B cell activation (Figure 3).

Helminth-induced immune modulation may provide benefits in mitigating severe outcomes of respiratory viral infections, but the resulting immune imbalance could have unintended consequences on vaccine efficacy for viral diseases. This raises the need for vaccines to incorporate adjuvants that compensate for helminth-induced immunosuppression, ensuring durable and effective protection in helminth-endemic countries. Helminth-derived molecules such as ES-62 and glycans offer valuable templates for synthetic adjuvant design, as they interfere with innate immune pathways like NF-κB and TLR signaling [43,44]. These mechanisms, while potentially dampening traditional vaccine responses, provide critical insights into developing novel adjuvants that can modulate immune responses in a more balanced manner. Previous studies have shown that rOv-ASP-1, a protein derived from the *Onchocerca volvulus* parasite, serves as an effective adjuvant, enhancing the efficacy of vaccines composed of proteins or synthetic peptides [45].

Mimicking helminth products like ES-62, which regulate inflammatory pathways, could enable the synthesis of vaccine molecules that enhance antigen-specific responses while simultaneously reducing adverse inflammation. This dual functionality is particularly advantageous in creating vaccines that are both effective and safer, with reduced side effects. Such approaches could include combining helminth-inspired adjuvants with immune stimulants, such as Toll-like receptor (TLR) agonists, to promote Th1 responses while the helminth-derived components mitigate excessive inflammation [46]. Alternatively, derivatives of helminth molecules such as glutathione S-transferase from *Fasciola hepatica* (nFhGST) could be synthesized to retain their ability to modulate NF-κB or TLR pathways while avoiding complete suppression of immune activation [43]. Structural modifications of helminth-derived molecules could also be employed to selectively retain their anti-inflammatory effects while reducing their capacity to suppress critical immune responses, such as Th1 or Th17 pathways. This strategy would ensure a balanced immune activation, preserving the ability to generate robust vaccine-induced immunity. The goal is to leverage the immunosuppressive capabilities of helminth-derived molecules in a selective manner, minimizing harmful inflammation without compromising the immune responses necessary for protection and vaccine efficacy. Advanced techniques, including molecular docking, bioinformatics, and synthetic biology, can play a pivotal role in achieving this delicate balance, paving the way for innovative and efficient vaccine designs.

The consensus among most studies points to the potential impact of helminth coinfection in altering the course of respiratory viral diseases, including SARS-CoV-2, RSV, and Influenza, through immune modulation and gut microbiome interactions [47], potentially leading to milder disease outcomes. This modulatory effect is likely driven by anti-inflammatory pathways, reduced pro-inflammatory responses, microbiome support, and the interruption of TLR and NF-κB pathways, suggesting a potential role for helminth-induced immunomodulation and a balanced gut microbiome in mitigating severe respiratory outcomes, although the extent of modulation varies with the type of helminth, the stage of helminth infection, and the specific respiratory virus involved. This highlights a complex interplay between helminthic infections and respiratory viruses, which warrants further investigation to fully understand the underlying mechanisms and implications for disease management.

**Table 1 vaccines-13-00436-t001:** Study characteristics of some included studies on helminth–respiratory virus interaction.

Study	Type of Study	Year of Publication	Helminth or Anthelmintic Treatment Involved	Respiratory Disease Condition Investigated	Main Findings ↓↑	Ref.
Helminth Infection and COVID-19 Severity in Ethiopia	Human	2021	Helminths and protozoa	COVID-19	↓ COVID-19 severity.	[48]
Helminth Antigens and SARS-CoV-2 Response	Human (In vitro)	2022	Helminth antigens	COVID-19	↓ SARS-CoV-2-reactive CD4+ T cells, ↑ IL-10, ↓ IFNγ and TNFα levels.	[30]
*H. polygyrus* and Influenza Virus	Mice	2023	*H. polygyrus bakeri* (Hpb)	Influenza	↓ weight loss, ↑ immune responses in the lungs, ↓ viral loads, and ↓ inflammation in lungs.	[34]
*H. polygyrus* Infection and RSV	Mice (BALB/c)	2017	*H. polygyrus*	Respiratory Syncytial Virus	↑ monocytes in the blood and lungs driven by IFN-I signaling, ↓ viral load.	[36]
Filarial Infection Stage and Influenza	Mice (BALB/c)	2022	Filarial infection	Influenza	early-stage infection reduced symptoms, middle-stage infections offered less protection, and late-stage infections worsened symptoms and ↑ viral loads.	[35]
*H. polygyrus* and RSV	Mice (BALB/c)	2024	*H. polygyrus*	Respiratory Syncytial Virus	↓ RSV severity, ↓ lung inflammation.	[37]
*Ascaris Lumbricoides* and COVID-19 Severity in Benin	Human	2023	*Ascaris lumbricoides*	COVID-19	↓ risk of severe COVID-19, ↓ systemic pro-inflammatory markers.	[29]
Helminth and SARS-CoV-2	Human	2025	*A. lumbricoides*, *S. ratti*, *A. viteae*	COVID-19	↓ COVID-19 severity, ↓ inflammatory cytokines, ↓ SARS-CoV-2 antibodies	[16]

This table represents the key findings of some selected studies on the interaction between helminth and respiratory virus infections. It also includes the type of study, year of publication, the main findings, and references. ↑ stands for increase; ↓ stands for decrease.

### 3.2. Helminths and Blood-Borne Viral Co-Infections

Co-infection of helminths with blood-borne viral infections, including Hepatitis B Virus (HBV), Human Papillomavirus (HPV), herpes simplex virus (HSV), Human T-lymphotropic Virus (HTLV), Vaccinia virus, Puumala hantavirus (PUUV), and Human Immunodeficiency Virus (HIV), have been studied in human and animal models. Contrary to the generally observed beneficial effect of helminths on respiratory virus infection outcome, their interaction with blood-borne viruses often reveals a detrimental relationship, leading to worsened clinical presentations, including increased viral loads, enlarged spleens, and more severe illness [49,50,51,52]. This exacerbation is often due to the helminths’ promotion of a type 2 immune response, which modulates the type 1 responses necessary for controlling viral clearance [53]. Few studies report no effect on disease outcome while very rare cases report a beneficial effect. In the context of human viral infections, helminth infections like those caused by *Ascaris* and *Trichuris* have been linked to higher viral loads and lower CD4+ counts in HIV-infected individuals, complicating HIV management [51]. Research on lymphatic filariasis (LF) and HIV co-infection reveals mixed findings. While few studies report no significant differences in LF prevalence or worm burden between HIV-positive and HIV-negative individuals [54], others highlight that lymphatic filariasis increases HIV susceptibility and incidence, particularly in individuals with microfilariae [55,56], suggesting that helminth-driven immune modulation increases vulnerability to viral infections. Deworming against helminth infections led to a decrease in serum IgE and improvement in antiviral immune response, viral clearance, and reduced viral severity in HIV patients [49]. Hookworm infection was also found to be associated with a heightened risk of sexually transmitted infections (STIs), bacterial vaginosis (BV), candidiasis, HPV, and viral load [57,58], likely due to Th2-biased immune responses. Similar trends were observed with soil-transmitted helminths (STH), which increased HPV prevalence in endemic regions. In the female reproductive tract, helminth-induced immune changes exacerbated HPV susceptibility and bacterial infections. Although some studies involving *Schistosoma mansoni* indicate a nuanced impact on HBV, showing improved immune responses during acute infection phases without worsening chronic HBV [59], most human research supports that helminth co-infections with blood-borne viruses generally worsen viral disease outcomes [50,60,61]. The findings in an animal model also demonstrated that mice co-infected with helminths (*Ascaris and Nippostrongylus braziliensis* (Nb)) and viruses (Vaccinia virus and HSV-2, respectively) exhibited more severe disease, increased viral loads, and reduced CD8+ T cells than those infected with the virus alone [62]. Mechanistic findings revealed that *Nippostrongylus braziliensis* (Nb) modulates the immune response in mice co-infected with HSV-2 by imprinting a type 2 immune signature in the female genital tract (FGT) [63]. Nb infection induced epithelial stress and the release of IL-33, which activated innate lymphoid cells (ILC2s) and inflammatory monocytes to produce IL-5, driving eosinophil recruitment and persistence in the FGT. This eosinophil influx exacerbated HSV-2-induced epithelial necrosis, genital pathology, and ulceration without altering viral replication [63]. The mechanism involved IL-5-driven eosinophilic inflammation, with raised levels of IL-5 and reduced antiviral IFN-γ responses impairing epithelial MHC expression and antiviral immunity. Depleting IL-5 reduced eosinophil infiltration and pathology, confirming its role. Notably, the exacerbated pathology is independent of IL-4 receptor alpha (Il4rα) signaling, highlighting the centrality of IL-5 and eosinophilic inflammation in Nb-enhanced HSV-2 pathogenesis [63]. Thus, helminth infections frequently complicate systemic viral disease management and underscore the need for integrated treatment strategies for co-infected individuals. Table 2 provides supporting information on the interaction between helminths and blood-borne viral infections.

Comparing disease outcomes in co-infections with respiratory versus blood-borne viruses reveals intriguing differences despite both being viral infections. Respiratory viruses, including recent SARS-CoV-2, often lead to outcomes heavily influenced by the level of inflammation, commonly referred to as a cytokine storm [64,65]. Effective management of these conditions focuses on controlling this inflammatory response [66,67,68], and helminth infections, which promote a Th2 immune response, might potentially mitigate the cytokine storm by suppressing the Th1 immune response, leading to better health outcomes [29,69]. However, this benefit comes with a trade-off, the suppression of Th1 immune response by helminths can suppress antibody production and the neutralization potential necessary for viral clearance. Despite this, the cytokine storm remains the major predictor of disease outcome, and even modest levels of antibodies are typically sufficient to clear the virus. In contrast, blood-borne viruses, such as HIV, HBV, HPV, and HTLV, exhibit a distinct pathogenic mechanism. Blood-borne retroviruses such as HIV and HTLV integrate their RNA genomes into the host DNA, resulting in chronic infections with persistent viral replication [70]. This integration disrupts normal cellular processes and causes ongoing immune system damage. HIV targets and destroys CD4+ T cells, which are crucial for an effective immune response, leading to severe immunodeficiency and an increased risk of opportunistic infections and cancers [71]. Additionally, retroviruses evade immune detection through rapid mutations and immune evasion strategies, further exacerbating disease severity. Consequently, the management of retroviral infections often does not focus on cytokine storm mitigation, as in respiratory viruses, but on viral clearance, which depends on enhanced Th1-mediated T cell and antibody responses. However, in the context of helminth co-infection with retroviruses, their suppressive effect on Th1 responses induces impaired antibody production and neutralization against systemic viruses, potentially worsening overall health outcomes rather than alleviating symptoms, as seen with respiratory viral infection [50,53,62].

In conclusion, while helminth co-infections may modulate the immune system, leading to better health outcomes in respiratory virus conditions like SARS-CoV-2, RSV, and Influenza by dampening inflammatory cytokine storms through skewed Th2 immune modulation, this same immune suppression can lead to worsened clinical outcomes in systemic viral infections, such as HIV, HTLV, HBV, HPV, and HSV, where the Th1 responses are essential for viral clearance and immune defense, thus potentially leading to poorer health outcomes and increased viral load (Figure 4).

**Figure 4 vaccines-13-00436-f004:**
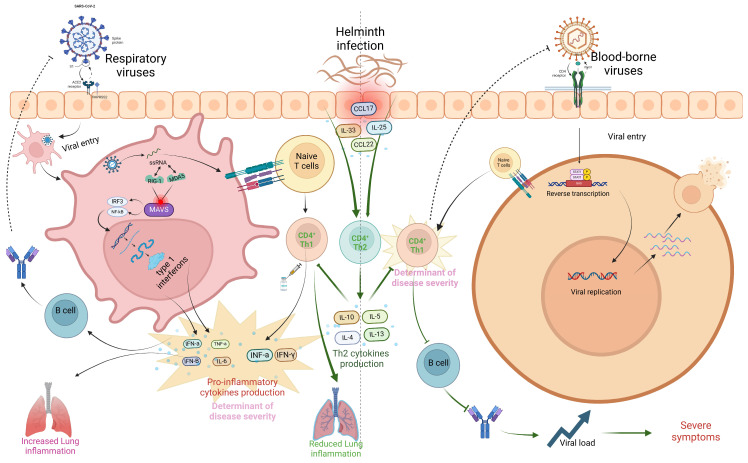
Influence of helminths on SARS-CoV-2 and HIV infection outcome. The virus enters the host cell via endocytosis after binding to the ACE2 receptor, facilitated by host proteases like TMPRSS2. Following fusion, viral RNA is released into the cytoplasm, where it is detected by pattern recognition receptors (PRRs) such as RIG-I and MDA5. This activates the MAVS signaling complex, which induces the production of type 1 interferons (IFN-α, IFN-β) and pro-inflammatory cytokines through NF-κB and IRF3/IRF7 pathways. Dendritic cells present viral antigens to naïve CD4+ T cells, promoting Th1 cell differentiation via IL-12. Th1 cells secrete IFN-γ, enhancing macrophage activity and CD8+ T cell activation. This creates a feedback loop of interferon production, which, if unregulated, can contribute to cytokine storms. Helminth co-infection suppresses Th1 cells, reducing cytokine production and mitigating the cytokine storm. In the case of HIV, the virus binds to the CD4 receptor, fuses with the host cell, and releases viral RNA. Reverse transcription converts viral RNA into DNA, which is integrated into the host genome. This leads to the production of new viral particles, which spread infection by targeting CD4+ T cells. Helminth co-infection further suppresses CD4+ T cells, reducing immune response and increasing viral load, leading to severe outcomes (created using BioRender.com).

**Table 2 vaccines-13-00436-t002:** Study characteristics of some included studies on helminth–systemic virus interaction.

Study	Type of Study	Year of Publication	Helminth or Anthelmintic Treatment	Systemic Viral Condition Investigated	Main Findings ↓↑	Ref.
Ascaris and Vaccinia Virus	Mice	2017	*Ascaris* infection	Vaccinia Virus (VACV)	↑ severity, ↑ viral loads, ↑ lung inflammation, ↓ CD8+ T cells.	[62]
Helminth Infection and HIV Immune Response	Human	2014	*A. lumbricoides*, *T. trichiura*	HIV	↑ CCR5 expression on CD4 or CD8 T cells, ↑ HIV susceptibility, ↓ HLA-DR+ CD8 T cells.	[61]
HIV and LF Interaction and treatment in Tanzania	Human	2016	*Wucheriria bancrofti*	HIV	No significant difference in LF prevalence or worm burden was observed between HIV-positive and HIV-negative individuals.	[54]
LF and HIV Incidence in Tanzania	Human	2016	*Wuchereria bancrofti*	HIV	HIV incidence was significantly higher among individuals with LF compared to those without.	[55]
LF and HIV Risk by Microfilariae in Tanzania	Human	2023	*Wuchereria bancrofti*	HIV	Individuals with microfilariae-producing *W. bancrofti* had a significantly higher HIV incidence than those without.	[56]
Hookworm and HPV Co-Infection	Human	2022	Hookworm (Ancylostoma duodenale)	HPV	Hookworm infection was associated with a mixed type 1/type 2 immune response in the vaginal tract and an increased risk and viral load of HPV infection.	[58]
Hookworm and FRTIs in Rural Togo	Human	2021	*Hookworm*	HPV and FRTIs (Female Reproductive Tract Infections)	Hookworm infection increased the risk of HPV infections and was associated with sexually transmitted infections (STIs), bacterial vaginosis (BV), and candidiasis.	[57]
STH and HPV in Peru	Human	2016	*STH*	HPV	Women with STH infections in the Peruvian Amazon had a 60% higher HPV prevalence compared to uninfected women. A Th2 immune profile was observed in cervical fluid of helminth-infected women.	[72]
Nippostrongylus *braziliensis* Infection Alters FGT Immunity	Mice	2021	Nippostrongylus *braziliensis*	HSV-2	Co-infection led to enhanced genital ulceration, necrosis, reduced MHC expression, and impaired IFN-γ response without affecting viral load.	[63]
Helminth and HIV	In-vitro	2016	Helminth proteins (BmA and ES-62)	HIV	Neither BmA nor ES-62 affected overall HIV-1 replication in CD4+ T cells or dendritic cell function.	[73]
*L. sigmodontis* and Retroviral Infection	Mice (C57BL/6)	2016	*L. sigmodontis*	Retroviral infection	↑ viral load, larger spleens, more severe illness, ↓ virus-neutralizing antibodies, ↑ viral loads.	[50]
Puumala hantavirus and Heligmosomum mixtum	Mice	2014	*Heligmosomum mixtum*	Puumala hantavirus (PUUV)	↓ immune responses, ↑ viral loads.	[60]
*S. mansoni* and HBV	Mice	2020	*Schistosoma mansoni*	Hepatitis B Virus (HBV)	↑ antiviral response to HBV, ↑ IFN-γ.	[59]
Helminth Infections in HIV-patients	Human	2015	Deworming	HIV	↓ symptoms, ↓serum IgE	[49]
Helminth Infections and Herpesvirus Interactions	Mice (C57BL6/)	2014	general helminth infections	Murine g-herpesvirus	↑ IL-4, ↑ viral replication and reactivation.	[74]
Schistosoma and Geohelminth Infections with β-cell Function and Insulin Resistance (Tanzania)	Human	2022	Schistosoma and Geohelminth Infections	HIV/Insulin	Schistosoma infection ↑ β-cell function in HIV-uninfected participants. In HIV-infected individuals not on ART, Schistosoma and geohelminth infections ↓ β-cell function, and caused insulin resistance.	[75]
Impact of Schistosomiasis Treatment on HIV Susceptibility in Women (Uganda)	Human	2019	*S. mansoni* infection and treatment	HIV	Schistosomiasis treatment ↓ HIV entry into cervical and blood CD4+ T cells, with effects lasting up to two months. The treatment led to immune activation but also ↑ IFN-I pathway activity, which potentially ↓ HIV susceptibility by inhibiting HIV entry.	[76]
Helminth-induced immunomodulation and its effect on antiviral immunity	Mice	2014	*Trichinella spiralis*	Murine Norovirus (MNV), CW3 strain	↓ antiviral immunity independently of changes in the microbiota. The impairment was mediated by STAT6-dependent alternative macrophage activation, with the molecule Ym1 contributing to ↓ antiviral response. Neutralization of Ym1 partially restores antiviral immunity.	[77]
HIV and helminth Co-Infections in China	Human	2013	Intestinal helminths and protozoa (*Blastocystis hominis*, *Cryptosporidium* spp.)	HIV	Co-infection with helminths led to a shift in Th1–Th2 balance similar to HIV infection, potentially accelerating AIDS progression.	[53]
Helminth-HIV Co-Infection in South Africa	Human	2011	*A. lumbricoides*, *Trichuris trichiura* (diagnosed via coproscopy and IgE levels)	HIV	Individuals with both helminth egg excretion and high *Ascaris*-specific IgE had dysregulated immune cells, higher viral loads, and increased immune activation. Those with egg excretion but low IgE had a modified Th2 response with better HIV-related immune profiles	[51]
Helminth Co-Infection and HTLV-1 Immune Modulation	Human	2005	*Strongyloides stercoralis*, *Schistosoma mansoni*	Human T cell Lymphotropic Virus Type 1 (HTLV-1)	HTLV-1 carriers co-infected with helminths had lower IFN-γ levels and reduced activation of CD8+ and CD4+ T cells compared to HTLV-1 carriers without helminths. IL-5 and IL-10 levels were higher in co-infected individuals	[52]

This table represents the key findings of some selected studies on the interaction between helminths and non-respiratory virus infections. It also includes the type of study, year of publication, the main findings, and references. ↑ stands for increase; ↓ stands for decrease.

### 3.3. Helminth and Bacterial Infections

Helminths and bacterial coinfections are often common, and studies have demonstrated the co-infection of helminths with bacterial infections in tropical and subtropical regions, affecting millions of people globally [78,79]. In many tropical and subtropical areas where helminths are highly prevalent, bacterial infections, such as those caused by *Mycobacterium*, *Salmonella*, or *Helicobacter pylori*, are also prevalent [79]. Coinfection can lead to complex interactions between the immune system and pathogens. Helminths often modulate the host’s immune response, skewing it toward an anti-inflammatory, Th2-dominated response, which can affect the body’s ability to effectively fight off infections, including bacterial infections, with unintended effects on vaccine efficacy [80]. The effect of helminth infections on bacterial diseases can vary widely depending on the type of bacterial pathogen, the nature of the helminth infection, and the immune status of the host. The interaction between helminth infections and bacterial pathogens, particularly *Mycobacterium species,* has been highly studied, and it reveals a multifaceted relationship that can influence disease outcomes in various ways. Most reviewed studies suggest that helminth co-infections may exacerbate bacterial infections by impairing crucial immune responses [80,81,82]. Studies on *Heligmosomoides polygyrus* co-infection with Salmonella in mice highlight how helminths can reduce neutrophil recruitment to the infection site, which is vital for controlling bacterial spread. This impaired immune response led to more severe intestinal inflammation and poorer overall outcomes, indicating that helminth infections may hinder the body’s ability to effectively combat bacterial pathogens [81]. However, the immunological basis of this association requires further expansion. Similarly, co-infection with *Schistosoma mansoni* or *Nippostrongylus brasiliensis* with mycobacterial infection in mice could reduce Mincle expression in macrophages, affecting immune responses to mycobacterial infections and vaccines [80]. Deworming with albendazole 400 mg/day for 3 days was shown to reduce eosinophil counts and IL-10 levels in helminth-TB co-infected patients, showing a reversal of helminth-induced immune suppression, implying that removing helminths can restore normal immune function and improve the host’s ability to control bacterial infections [83]. This finding supports the idea that in some cases, helminth infections may need to be addressed as part of a broader strategy to manage bacterial diseases effectively. Additionally, the impact of maternal helminth infections on neonatal immunity adds another layer of complexity. Studies on the human population show altered immune responses in newborns born from helminth-infected mothers, such as higher IgE levels and modified TB-specific antibody responses, suggesting that maternal helminth infections can have long-lasting effects on the offspring’s immune system [82]. These alterations could influence how effectively vaccines work and how the infant’s immune system responds to infections later in life.

Conversely, few studies observed that helminth infections might mitigate disease severity in certain contexts, involving some bacterial infections. Chronic *L. sigmodontis* infection has been found to improve sepsis outcomes caused by *E. coli* by reducing inflammation and enhancing bacterial clearance, and it also affects macrophage function through Wolbachia and TLR2 signaling [84]. *Strongyloides stercoralis* infection was also associated with reduced TB severity in some patients [85]. However, more investigations are needed to unravel the immunological basis of this association. This suggests that the immunomodulatory effects of helminths, which typically induce a Th2-dominated immune response, might help mitigate the immune response to some bacterial infections, potentially leading to less severe disease [86]. These findings are consistent with the notion that in chronic bacterial infections, where inflammation can cause significant tissue damage, the anti-inflammatory effects of helminths might be beneficial [87].

While helminth infections may often worsen bacterial disease outcomes by impairing essential immune responses necessary for bacterial clearance, they may also mitigate severity in some scenarios by reducing harmful inflammation in some chronic infections. The variability in outcomes underscores the importance of in-depth and future research exploring the interaction between helminths and bacterial infection that may be context-dependent, including the type of helminth, the specific bacterial infection, and the host’s immune status, as well as the importance of maternal health in shaping neonatal immune outcomes. Therefore, tailored approaches, including the strategic use of deworming, may be necessary to optimize disease management in populations where helminth and bacterial co-infections are prevalent. Table 3 provides supporting information on the interaction between helminths and bacterial infections.

### 3.4. Helminths and Non-Communicable Diseases

Non-communicable diseases (NCDs), such as cardiovascular diseases, diabetes, cancer, and chronic respiratory diseases, are major global health challenges, particularly in low- and middle-income countries [91,92]. These conditions are primarily driven by lifestyle factors like poor diet, lack of physical activity, smoking, and environmental factors. In many endemic regions where helminths are prevalent, there is also a growing concern of NCDs, especially diabetes. Infection with parasitic worms like roundworms and hookworms can affect the progression and management of NCDs. While they may reduce inflammation associated with NCDs, they can also impair the host’s ability to regulate metabolic processes and fight chronic diseases effectively. Understanding the interplay between helminths and NCDs is crucial for developing integrated health strategies in regions where both conditions coexist. The interaction between helminth infections and non-communicable diseases reveals a complex dynamic that can have both beneficial and detrimental outcomes, depending on the specific helminth and the context of the infection. Several studies suggest that co-infections with helminths may offer some benefits in managing certain NCDs by modulating the immune system.

In metabolic disorders like type 2 diabetes (T2D), helminth infections and derived products appear to offer some benefits in managing such conditions. Studies on *H. polygyrus* showed improved glucose control, reduced insulin resistance, and fat accumulation in diabetic mice, suggesting that helminth-induced immune modulation could be a promising therapeutic approach for managing T2D [93] (Figure 5). Furthermore, filarial infections have been linked to reduced inflammation and improved insulin sensitivity, with regions experiencing an increase in T2D incidence following the elimination of filarial diseases [94,95]. This suggests that chronic helminth infections may play a role against the development of metabolic dysfunctions by regulating inflammation.

In terms of allergic conditions, helminths have shown potential in controlling allergic responses. Early-life infection with *H. polygyrus* in mice led to chronic mild inflammation but a controlled allergic response later in life, supporting the idea that helminth infections can regulate immune responses and reduce the likelihood of allergic inflammation [96]. Similarly, *N. brasiliensis* infections reduced allergic responses, particularly when the infection occurred weeks before allergen exposure, with the protective effect linked to IL-10 [97]. Chronic infection with *Schistosoma mansoni* also induces a type 2 immune response characterized by elevated interleukin-10 (IL-10) production, which modulates both type 1 and type 2 immune responses, reducing the risk of autoimmune diseases like multiple sclerosis, diabetes, and Crohn’s disease, as well as allergic conditions such as asthma [98,99,100]. The hygiene hypothesis suggests that helminths suppress bystander immune responses to allergens, explaining the lower prevalence of asthma in developing regions compared to industrialized countries. Epidemiological and experimental studies demonstrate an inverse association between schistosomiasis and allergy severity, with reduced asthma pathology and poor skin prick test reactivity in endemic areas, and protection was linked to IgE production [101]. In animal models, schistosome infections suppress allergic airway inflammation (AAI) when induced during the patent phase of infection, which involves egg production by fecund female worms [102]. This suppression is mediated by infection-induced CD4+Foxp3+ regulatory T cells (Tregs), as their depletion reverses the protective effect, leading to aggravated AAI. The suppression effect is lost after anti-helminthic treatment, likely due to delayed Treg reconstitution. These findings suggest that helminth-derived products, particularly during the egg-producing phase, could be harnessed to develop therapies targeting Treg induction for managing immune-mediated diseases like asthma [102]. The modulatory effect of helminths in allergy and autoimmune diseases extends beyond the direct impact on the host; the offspring of mice mothers infected with *Schistosoma mansoni* during specific immune phases exhibited a reduced likelihood of developing allergic airway inflammation, highlighting the potential long-term protective effects of maternal helminth infections [103]. In Gabonese mothers, placentas showed a significantly lower gene expression of VDR, Foxp3, IL-10, and Cyp27b1 compared to German mothers (non-endemic), suggesting that prenatal helminth exposure may impair immune system development, leading to altered immune responses in offspring [104]. These findings align with the “hygiene hypothesis”, which suggests that decreased exposure to infections may increase susceptibility to allergic diseases.

Helminth infections have also shown promise in treating autoimmune diseases and neurodegenerative conditions. *Trichinella spiralis* modulates immune responses to favor a Th2 and regulatory environment, potentially reducing inflammation and providing therapeutic benefits for autoimmune conditions [105]. In neurodegenerative diseases, co-infection with *H. polygyrus* reduces prion accumulation and extends survival in mice with prion disease, indicating that helminth-induced immune modulation could slow disease progression [106]. However, helminth infections are not without risks. While they may offer immune protection and modulation, clearing the infection can reverse these benefits, as seen in *Strongyloides stercoralis* infections in people with T2D. Once the infection was treated, the reduced inflammation and microbial leakage benefits were lost [107], raising concerns about the long-term management of such co-infections. Furthermore, chronic mild inflammation induced by helminths may have unintended consequences, particularly if it persists or interacts with other inflammatory conditions.

Overall, co-infections with helminths appear to offer a range of benefits for managing NCDs, particularly in regulating immune responses to reduce inflammation, improve metabolic outcomes, and control allergic and autoimmune conditions (Figure 5). However, the potential risks, particularly in terms of chronic inflammation or the loss of benefits following helminth clearance, must be carefully considered. These suggest that helminth-based therapies or controlled immune modulation could be a valuable tool in managing NCDs, though further research is needed to balance the risks and benefits of such approaches. Table 4 provides supporting information on the interaction between helminths and non-communicable diseases.

### 3.5. Helminths and Vaccine Efficacy

Vaccines are biological preparations designed to provide immunity against infectious diseases. They contain antigens, derived from or mimicking pathogens, which stimulate the immune system to recognize and fight infections. The primary purpose of vaccines is to “train the immune system” to develop a memory of the pathogen, allowing for a faster and more effective response upon future exposure. The basic mechanism of action involves the activation of the body’s adaptive type 1 immune response, leading to the production of antibodies by B cells and the activation of T cells to destroy infected cells. Vaccines typically elicit a humoral immune response (antibody production) and a cell-mediated immune response (T cell activation). However, the efficacy of vaccines can be influenced by coinfections, including infections with helminths. The interaction between helminths and vaccine efficacy reveals a complex picture where helminths can both impair and sometimes have negligible effects on vaccine responses. Evidence from various reviewed studies indicates that helminth infections generally tend to be detrimental to vaccine efficacy. In mice infected with *Schistosoma mansoni*, the TD158 HIV-1 vaccine’s immune response was significantly weakened, suggesting that helminth infections can reduce the effectiveness of vaccines targeting HIV-1 [115]. Similarly, chronic infection with *Schistosoma japonicum* decreased the immune response to the HBV vaccine, though deworming with praziquantel improved this response, indicating that helminth infections can compromise vaccine efficacy but may be mitigated with appropriate treatment [20].

In the case of the BCG vaccine, deworming before vaccination enhanced immune responses, evidenced by increased levels of IFN-γ and IL-12. This indicates that helminths can decrease BCG vaccine efficacy by promoting an immunosuppressive environment through elevated TGF-β levels [17]. Similarly, *Heligmosomoides polygyrus* infection impaired the immune response to the Pfs25 DNA malaria vaccine. Although it did not affect the irradiated sporozoite vaccine, deworming improved responses to the Pfs25 vaccine, reinforcing the idea that helminth infections can diminish the efficacy of certain malaria vaccines but that this can be addressed through deworming [116]. For Influenza vaccines, mice with helminth infections showed impaired antibody responses to both seasonal and H1N1 Influenza vaccines. This impairment was associated with IL-10-producing regulatory T cells. Blocking IL-10 was found to partially restore vaccine responses, suggesting that targeting IL-10 might improve vaccine effectiveness in individuals with helminth infections [117]. The impact of helminths on COVID-19 vaccines has also been significant. Infection with helminths including *Trichinella spiralis*, *Schistosoma mansoni*, and *Heligmosomoides polygyrus* was found to reduce the efficacy of the COVID-19 mRNA vaccine, as evidenced by lower levels of antibodies, fewer splenic germinal center B cells, and lower T cell response [118,119,120]. Treatment with albendazole partially reversed these effects, indicating that chronic helminth infections can impair vaccine responses and deworming might help restore some level of efficacy [120]. However, deworming might not restore vaccine efficacy in all instances. Chronic worm infections were found to completely suppress the immune response to DNP-KLH and NIP-Ficoll vaccines, reducing B cells and antibody levels. Even after deworming, immune suppression persisted, requiring stronger vaccination strategies [13].

On the other hand, some vaccines have demonstrated resilience to helminth infections. The HPV-16/18 vaccine demonstrated high antibody levels in all participants regardless of malaria or helminth infections, and malaria-infected participants even showed slightly higher antibody levels, though the reasons for this observation are not yet clear [121]. Moreover, chronic helminth infections did not affect the effectiveness of the malaria vaccine, with similar immune responses observed in both infected and uninfected mice [122]. Furthermore, while maternal helminth infections initially led to lower antibody levels against tetanus in cord blood, vaccine responses in infants at 9-12 months were similar to those of infants born from uninfected mothers [123].

Overall, while few vaccines can retain effectiveness in the presence of helminth infections, the general trend indicates that helminth infections weaken vaccine-induced immune responses, particularly through mechanisms such as the increased production of immunosuppressive cytokines and disruptions in immune balance. This underscores the need for targeted strategies, such as pre-treatment with deworming medications or tailored vaccination approaches, to optimize vaccine protection in populations affected by helminths. Table 5 provides supporting information on the interaction between helminths and vaccine efficacy.

## 4. Conclusions

The outcome of helminth co-infection is highly diverse and context-specific. It could be either beneficial or detrimental. This variability depends on several factors, including the type and stage of helminth infection, the host’s immune status, and the pathogen involved in the co-infection. Helminths can sometimes reduce inflammation and improve outcomes in some bacterial and viral infections, like respiratory viruses including SARS-CoV-2, RSV, and Influenza, as well as bacterial sepsis by *E. coli*. They could also be beneficial in managing some non-communicable diseases such as T2D, asthma, and metabolic disorders. Still, they may also exacerbate illness in other cases, increasing viral loads or impairing immune responses. Additionally, helminth co-infection typically diminishes vaccine efficacy, with long-lasting effects even after deworming, particularly in maternal infections affecting neonatal immunity and vaccine response.

## 5. Limitations

Many studies included in this review are regionally biased, primarily conducted in areas where helminth infections are endemic, such as low- and middle-income countries. This focus on specific regions may limit the generalizability of conclusions, as immunological responses and environmental factors differ significantly across geographic locations. Methodological inconsistencies further complicated synthesis, with variations in diagnostic techniques, helminth species, and outcome measures. Some studies emphasized clinical outcomes, while others focused on immunological biomarkers, which may not always directly correlate. Moreover, different studies utilized different guidelines to score disease outcome/severity, which may bring some inconsistencies in the synthesis of the results. Additionally, the diversity in the study designs ranging from observational studies and cross-sectional analyses to animal models limits the robustness of comparisons, as each approach has distinct limitations. Observational and cross-sectional studies are informative but lack causative insights, while animal models may not fully replicate human immune responses to co-infections. A scarcity of longitudinal data was also notable; without long-term studies, it is challenging to understand the sustained effects of helminth co-infections on disease progression. These limitations underscore the importance of future studies with standardized methodologies, broader geographic representation, and longitudinal tracking to better understand the nuanced effects of helminth co-infections across various disease outcomes.

## 6. Recommendation and Future Research Directions

Routine deworming programs utilizing anthelmintics can significantly reduce the burden of helminth infections and restore immune function before vaccination. Therefore, developing vaccination programs that specifically account for helminth infections is crucial. However, it is important to recognize that the immunomodulatory effects of helminths may persist even after deworming, leading to a lingering state of immune suppression whose duration remains largely unclear in the human population. This uncertainty underscores the necessity for further research aimed at clarifying how long the immunosuppressive effects of helminths last. Such studies are crucial for determining the optimal timing for vaccination following anthelmintic treatment, as understanding the duration of immune suppression can help enhance vaccine efficacy and overall health outcomes in populations affected by helminth infections.

## Figures and Tables

**Figure 1 vaccines-13-00436-f001:**
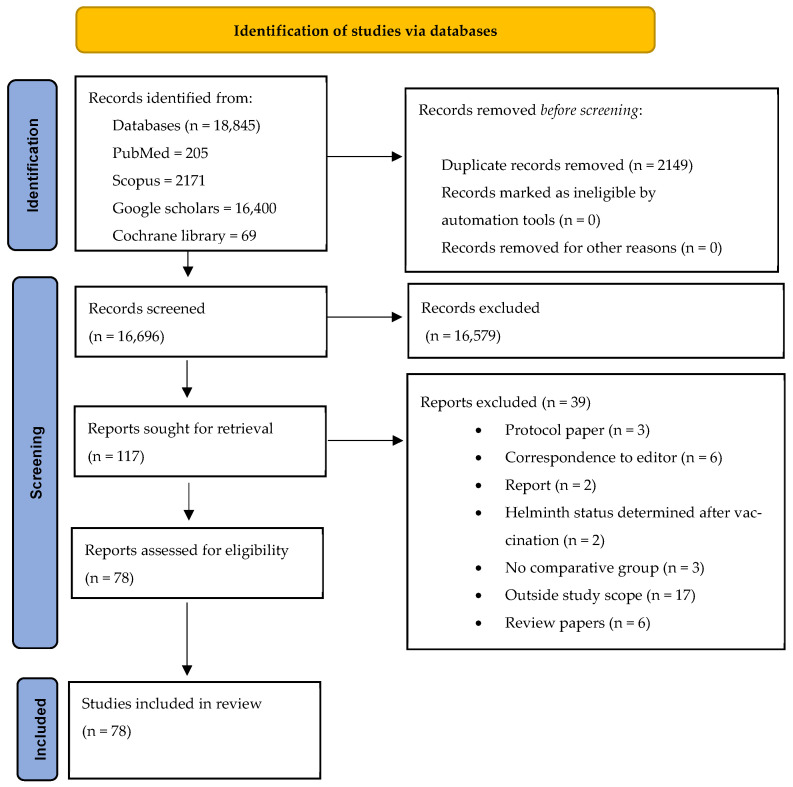
PRISMA flowchart.

**Figure 2 vaccines-13-00436-f002:**
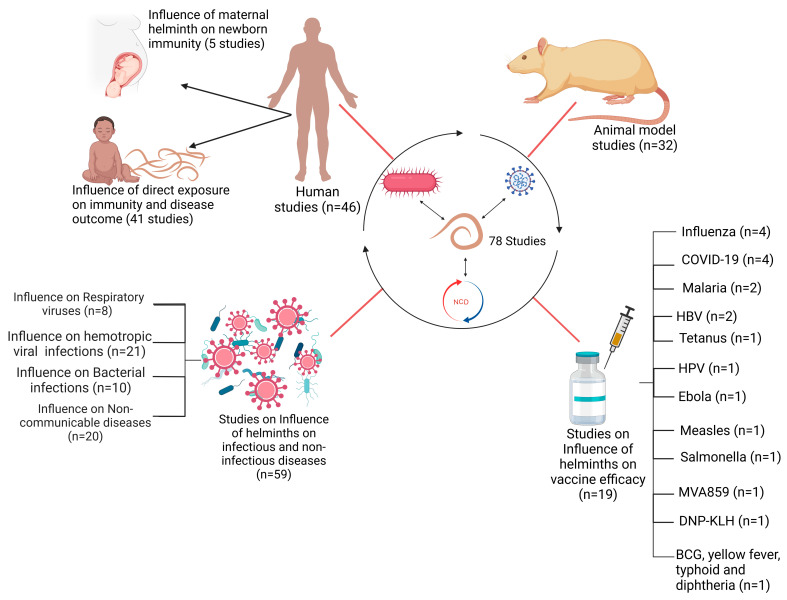
Schematic representation of studies included in this review.

**Figure 3 vaccines-13-00436-f003:**
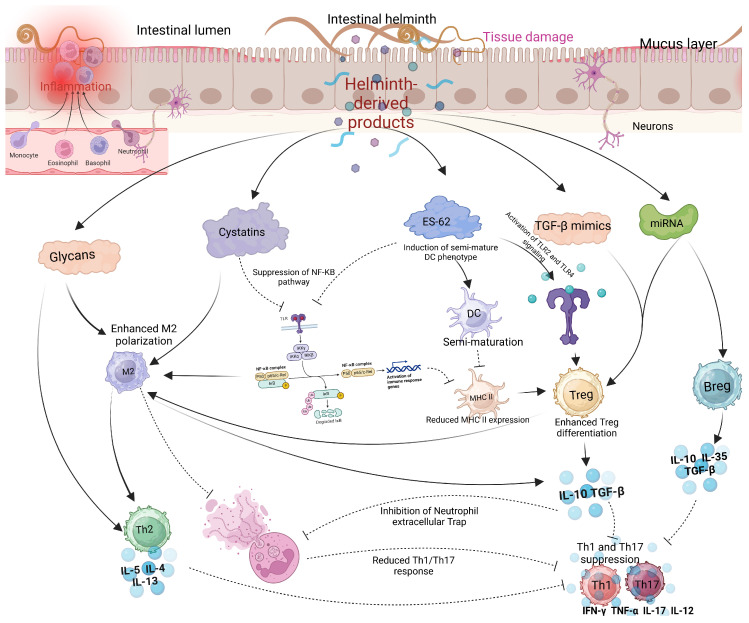
Modulatory effects of helminth-derived products on immune cells. Helminths produce several immunomodulatory molecules, including glycans, cystatins, ES-62, TGF-β mimics, and miRNAs. Glycans and cystatins promote M2 macrophage polarization, while cystatins and ES-62 suppress the NF-κB pathway, reducing the activity of Th1 and Th17 cells. Additionally, TGF-β mimics and miRNAs enhance the activation of Tregs and Bregs, further inhibiting Th1 and Th17 responses (created using BioRender.com).

**Figure 5 vaccines-13-00436-f005:**
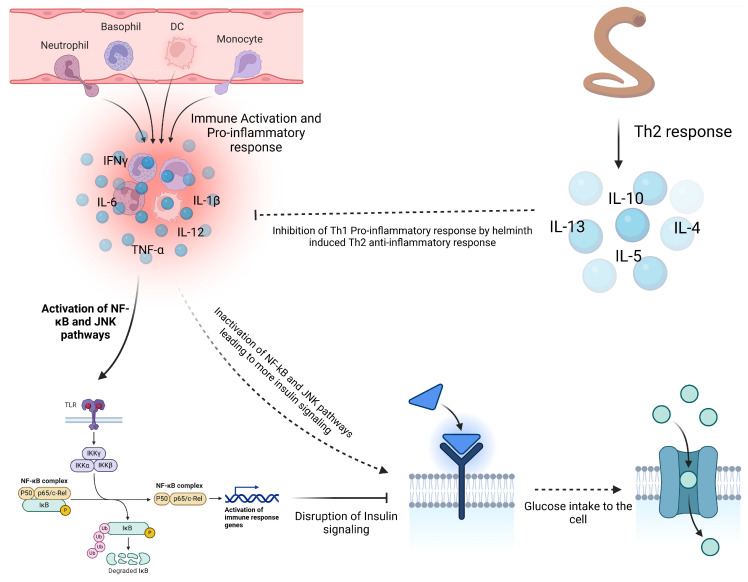
Influence of helminth on insulin resistance. Chronic inflammation in obesity contributes to the development of type 2 diabetes (T2D) by promoting insulin resistance. Pro-inflammatory cytokines like TNF-α and IL-6 activate pathways such as NF-κB and JNK, disrupting insulin signaling and reducing glucose uptake. In obese individuals, macrophages infiltrate adipose tissue, further enhancing inflammation and worsening insulin resistance. Co-infection with helminths helps mitigate this inflammation through Th2 anti-inflammatory cytokines, which inactivate the NF-κB and JNK pathways, restoring insulin signaling and glucose uptake and reducing T2D risk (created using BioRender.com).

**Table 3 vaccines-13-00436-t003:** Study characteristics of some included studies on helminth–bacterial interaction.

Study	Type of Study	Year of Publication	Helminth or Anthelmintic Treatment	Bacterial Disease Condition Investigated	Main Findings ↓↑	Ref.
*Heligmosomoides polygyrus* and *Salmonella Typhimurium*	Mice	2014	*Heligmosomoides polygyrus*	Salmonella Typhimurium	↑ intestinal inflammation, ↓ neutrophil, ↓ MIP-2, ↓ CXCL1, poor control of bacterial replication.	[81]
Modulation of Cytokine Responses in Latent Tuberculosis by Helminth Coinfection	Human	2017	Anthelmintic treatment (for Strongyloides stercoralis)	Latent TB	Individuals with latent TB and S. stercoralis co-infection had ↓ type 1 (IFN-γ, TNF-α, IL-2) and type 17 (IL-17A, IL-17F) cytokines but ↑ type 2 (IL-4, IL-5) and regulatory (TGF-β) cytokines. Anthelmintic therapy reversed these effects, increasing type 1 and type 17 cytokines while decreasing type 2 and regulatory cytokines. TB antigen-stimulated type 1 cytokines increased post-treatment.	[88]
IL-4, Helminths, and Mycobacterial Infections	Mice	2023	*Nippostrongylus brasiliensis*, *Schistosoma mansoni*	Mycobacterial infections (including BCG vaccine)	↓ Mincle expression in macrophages, affecting immune responses to mycobacterial infections and vaccines. IL-4 decreases Mincle activation and alters T cell responses depending on adjuvants.	[80]
Litomosoides sigmodontis and Bacterial Sepsis	Mice	2015	*Litomosoides sigmodontis*	Bacterial sepsis caused by *Escherichia coli*	Improves sepsis outcomes, ↑ bacterial clearance, ↓ pro-inflamatory cytokines. It also affects macrophage function through Wolbachia and TLR2 signaling.	[84]
Maternal Helminth Infections and TB	Human	2014	*S. mansoni*	Tuberculosis (TB)	↑ IgE levels and altered TB-specific antibody responses in newborns.	[82]
Helminth Infections and Tuberculosis Inflammation	Human	2014	*S. stercoralis*	Tuberculosis (TB)	↓ inflammation and immune activation markers in people with active TB, ↓ disease severity, ↓ acute phase proteins, ↓ matrix metalloproteinases, ↓ tissue inhibitors of matrix metalloproteinases, and ↓ sCD14 and ↓ sCD163.	[85]
Helminth co-infection in TB Patients in East Java	Human	2020	*Trichuris trichiura*	Tuberculosis (TB)	A total of 56% of active TB patients had *Trichuris trichiura* eggs, though were asymptomatic. Impact on TB diagnosis and treatment is unclear, suggesting further research is needed.	[78]
Helminth–Malaria–HIV co-infections and Latent TB Risk	Human	2014	Various helminths	Latent TB	Co-infections with helminths, malaria, or HIV did not increase the risk of latent TB infection or significantly affect the immune response in people with LTBI.	[89]
Helminth Modulation of Monocyte Responses in Latent Tuberculosis Infection (LTBI)	Human	*2020*	*S. stercoralis* infection and Anthelmintic treatment	Latent Tuberculosis Infection (LTBI)	Helminth co-infection linked to ↓ monocyte function, ↑ M2 polarization, and ↓ monocyte activation, but anthelmintic treatment reversed these effects after 6 months.	[90]
Helminth–tuberculosis co-infection in Ethiopia	Human	2015	Albendazole (400 mg/day for 3 days)	Tuberculosis	Albendazole treatment ↓ eosinophil counts and IL-10 levels in helminth-TB co-infected patients, showing a reversal of helminth-induced immune suppression. However, there was no significant improvement in clinical TB outcomes	[83]

This table represents the key findings of some studies selected studies on the interaction between helminth bacterial infections. It also includes the type of study, year of publication, the main findings, and references. ↑ stands for increase; ↓ stands for decrease.

**Table 4 vaccines-13-00436-t004:** Study characteristics of included studies on helminth–non-communicable disease interaction.

Study	Type of Study	Year of Publication	Helminth or Anthelmintic Treatment	NCD Investigated	Main Findings ↓↑	Ref.
Schistosomiasis and allergy susceptibility	Mice (BALB/c)	2014	*Schistosoma mansoni* infection	Allergies	Offspring of mothers infected with *S. mansoni* during certain immune phases (regulatory) had a **↓** chance of developing allergic airway inflammation. The protective effect is linked to maternal immune responses, not just the presence of the helminth infection.	[103]
*Heligmosomoides polygyrus* and type 2 diabetes	Mice	2016	*Heligmosomoides polygyrus* infection	Type 2 diabetes (T2D)	Infection with *H. polygyrus* improved glucose control, **↓** insulin resistance, and fat accumulation in diabetic mice, suggesting that helminth-induced immune responses might help manage T2D.	[93]
Early-Life heligmosomoides infection and allergy	Mice	2023	*Heligmosomoides polygyrus* infection	Allergic diseases	Early-life infection with *H. polygyrus* led to chronic mild inflammation and controlled allergic responses later in life. The mechanism involved IL-4 and FcγRIIb, suggesting that helminth infections could regulate allergies through immune modulation.	[96]
Strongyloides stercoralis and T2DM	Human	2020	*Strongyloides stercoralis* infection	Type 2 diabetes mellitus (T2DM)	In people with T2DM, *S. stercoralis* infection **↓** inflammation and microbial leakage. After treatment to clear the infection, these protective effects were reversed, suggesting that the helminth infection had a beneficial impact on managing T2DM.	[107]
Helminth infection and prion disease	Mice	2019	*Heligmosomoides polygyrus* infection	Prion disease	Co-infection with *H. polygyrus* **↓** early prion accumulation in Peyer’s patches and extended survival in mice with prion disease, indicating that helminth infections could slow the progression of prion diseases by modulating the immune response.	[106]
Helminth and skin inflammation	Mice	2017	*Heligmosomoides polygyrus* infection	Skin inflammation (allergic diseases)	*H. polygyrus* infection **↓** skin inflammation in mice by inducing an immune response that **↑** regulatory T cells, suggesting potential for helminth-based therapies in treating allergic diseases, including skin conditions like AD.	[108]
Therapeutic potential of *T. spiralis* AES in colitis	Mice (C57BL/6)	2014	Excretory/secretory products from *Trichinella spiralis*	Colitis (induced by DSS)	*T. spiralis* AES treatment significantly **↓** the severity of DSS-induced colitis in mice. The treatment led to improved inflammation, **↑** regulatory cytokines (IL-10, TGF-β), and regulatory T cells, while **↓** pro-inflammatory cytokines (IFN-γ, IL-6, IL-17).	[105]
Anthelmintic treatment and insulin resistance	Human	2017	Albendazole treatment	Insulin resistance (IR)	Albendazole significantly **↑** IR in helminth-infected subjects despite **↓** STH prevalence, total IgE, and eosinophil count.	[109]
Anthelmintic treatment and cardiometabolic risk	Human	2020	Praziquantel, Albendazole treatment	Cardiometabolic risk (IR, lipid profile, blood pressure)	Intensive treatment had no effect on IR but **↑** LDL cholesterol. Helminth infections (e.g., *Schistosoma mansoni*, *Strongyloides*) associated with **↓** LDL cholesterol, total cholesterol, triglycerides, and blood pressure.	[110]
*S. stercoralis* infection and obesity	Human	2020	*S.stercoralis* infection and anthelmintic treatment	Obesity and metabolic parameters	Helminth infection associated with **↓** insulin, GLP-1, and inflammatory cytokines in obese individuals. After treatment, insulin, GLP-1, and inflammatory cytokines **↑**, reversing the protective effect of helminths.	[111]
Prenatal helminth treatment and atopic diseases (entebbe mother and baby study)	Human	2017	Albendazole, Praziquantel (during pregnancy and early life)	Asthma, eczema, allergies (atopic diseases)	Prenatal and early-life helminth treatment had no significant impact on wheezing, asthma, or allergy risk at school age. Early eczema rates increased but did not translate to higher asthma rates later.	[112]
Hookworm infection and type 2 diabetes	Human	2023	*Necator americanus* larvae	Insulin resistance, type 2 diabetes	Hookworm infection **↓** fasting glucose, improved insulin resistance, and **↓** body mass in individuals at risk of type 2 diabetes. The infection was associated with gastrointestinal symptoms but was generally safe, indicating potential metabolic benefits.	[113]
Placental gene expression in helminth-endemic and non-endemic areas	Human	2019	*Schistosoma haematobium*	Immune modulation and allergy development	In Gabon (helminth-endemic), placentas showed significantly lower gene expression of VDR, Foxp3, IL-10, and Cyp27b1 compared to Germany (non-endemic), suggesting that prenatal helminth exposure may impair immune system development, leading to altered immune responses in offspring.	[104]
Intensive anthelmintic mass drug administration (MDA) on allergy-related diseases (Uganda)	Human	2019	MDA with Praziquantel and Albendazole	Allergy-related diseases (wheezing, skin prick test positivity, allergen-specific IgE)	While intensive deworming reduced helminth infection intensity, it did not significantly affect allergy-related outcomes.	[114]
Patent infections of *S. mansoni* and allergic airway inflammation	Mice	2013	*Schistosoma mansoni* infection and praziquantel (PZQ) treatment	Allergic airway inflammation (AAI)/asthma	*S. mansoni* infection during the patent phase reduced airway inflammation, eosinophil infiltration, and Th2 responses in allergic mice. Protection was lost when infection was treated with PZQ, suggesting helminth-mediated immunomodulation.	[102]
Asthma and helminth infection in Brazil	Human	2003	*S. mansoni* infection (natural infection)	Bronchial asthma	*S. mansoni* infection was associated with a lower prevalence of asthma attacks and reduced use of asthma medication. Subjects from helminth-endemic areas had lower histamine release levels and a lower frequency of positive skin prick test (SPT) responses compared to non-infected individuals.	[98]
*Schistosoma mansoni* infection and airway inflammation in mice	Mice	2007	*S. mansoni* infection (acute, intermediate, and chronic phases)	Asthma (allergic airway inflammation)	Chronic S. mansoni infection reduced OVA-induced airway eosinophilia, peribronchial inflammation, goblet cell hyperplasia, and airway hyperreactivity (AHR). Suppression was associated with IL-10 production and increased infection intensity.	[100]
SEA treatment and Foxp3 Treg in the pancreas of NOD mice	Animal (NOD mice)	2009	*S.mansoni* egg antigen (SEA) treatment	Type 1 Diabetes	SEA treatment prevents diabetes in NOD mice by increasing Foxp3+ Treg cells in the pancreas. SEA induces TGF-β, IL-10, IL-4, and IL-35 expression, creating an immunoregulatory environment. The protective effect is dependent on CD25+ Treg cells, and SEA directly promotes Foxp3+ Treg differentiation.	[99]
Circulating filarial antigen levels and anti-filarial antibody titer among type-2 diabetes subjects	Human	2010	lymphatic Filariasis (LF) infection	Type 2 Diabetes (T2D)	Lower CFA levels in diabetic individuals compared to normoglycemic individuals. Lower anti-filarial IgG levels in diabetic individuals. Diabetics had elevated IL-6 and GM-CSF levels, which were reduced in individuals co-infected with LF. IFN-γ was elevated in diabetics but unaltered by LF status. TGF-β was higher in diabetics compared to controls.	[94]
Filariasis and type 1 diabetes	Human	2010	Lymphatic Filariasis (LF)	Type 1 Diabetes (T1DM)	Individuals with T1DM had a significantly lower prevalence of filarial-specific IgG4 (2%) compared to non-diabetic individuals (14%) (*p* < 0.001). No significant difference was found in general antifilarial IgG levels, suggesting reduced active infection but similar exposure rates between groups. Socioeconomic factors were not confounding variables in the observed differences.	[95]

This table represents the key findings of some selected studies on the interaction between helminths and non-communicable diseases. It also includes the type of study, year of publication, the main findings, and references. ↑ stands for increase; ↓ stands for decrease.

**Table 5 vaccines-13-00436-t005:** Study characteristics of some included studies on helminth–vaccine efficacy interaction.

Study	Study Type	Year of Publication	Helminth Used for Infection	Vaccine Tested	Main Findings ↓↑	Ref.
Malaria and helminth infections on HPV vaccine (Tanzania)	Human	2014	Various helminths and malaria	HPV-16/18 Vaccine	The HPV-16/18 vaccine was effective in all participants, with high antibody levels. Malaria or helminth infections did not **↓** vaccine effectiveness; malaria-infected participants even showed slightly higher antibody levels, though the reason is unclear.	[121]
Chronic helminth infection in malaria vaccine	Mice	2019	*Heligmosomoides polygyrus bakeri* (Hpb)	Malaria vaccine	Chronic helminth infection did not **↓** the effectiveness of the malaria vaccine. Both infected and non-infected mice showed similar immune responses and vaccine effectiveness, indicating that the vaccine remains effective even in the presence of helminth infections.	[122]
Helminth infection on COVID-19 vaccine	Mice	2024	Hpb	mRNA COVID-19	The vaccine produced similar B cell responses in both infected and uninfected mice. However, T cell responses were significantly weaker in Hpb-infected mice, leading to reduced protection against the Omicron variant. The suppression was linked to IL-10, and blocking IL-10 improved T cell responses.	[118]
Helminth infection on influenza vaccine	Mice	2019	*Litomosoides sigmodontis*	Influenza vaccine	Mice with helminth infections showed **↓** antibody responses to both seasonal and H1N1 Influenza vaccines. The impairment was linked to IL-10-producing regulatory T cells. Blocking IL-10 partially restored the vaccine response.	[117]
Helminth infection on oral and parenteral vaccines	Mice	2023	Hpb	Salmonella vaccine	Helminth infection **↓** the immune response to both oral and injected vaccines by disrupting the balance of regulatory T cells in the gut. This led to weaker Th1 and Th2 responses, suggesting the need for the pre-treatment of helminth infections in vaccine campaigns.	[124]
COVID-19 vaccine and *S. mansoni*	Mice	2023	*S. mansoni*	mRNA COVID-19 vaccine	Infected group that had received COVID-19 vaccine showed **↑** IFN-γ and TNF-α levels and **↓** levels of IL-4 and IL-17	[119]
*Litomosoides sigmodontis* on vaccine efficacy in mice	Mice (BALB/c)	2014	*Litomosoides sigmodontis*	DNP-KLH and NIP-Ficoll	Chronic worm infections completely suppress the immune response to the vaccine, reducing B cells and antibody levels. Even after deworming, immune suppression persisted, requiring stronger vaccination strategies.	[13]
*S.mansoni* on hepatitis B vaccine	Human	2023	*S.mansoni*	HBV vaccine	Higher *S. mansoni* worm burden correlated with **↓** HepB vaccine titers and altered immune responses, including **↓** circulating T follicular helper (cTfh) cells, **↑** regulatory T cells (Tregs), and changes in cytokine/chemokine levels. These immune alterations contributed to a blunted vaccine response.	[125]
*Litomosoides sigmodontis* on Influenza vaccine	Mice	2021	*Litomosoides sigmodontis* and Flubendazole (FBZ)	Influenza vaccine	Deworming improved vaccine protection only partially, highlighting that immune suppression persisted after deworming. Enhanced vaccination strategies were needed for full protection.	[126]
Maternal helminth infections on newborn vaccine responses	Human	2020	Maternal helminth infections during pregnancy	Tetanus vaccine	Babies born to mothers with helminth infections had **↓** antibody levels against tetanus in cord blood but showed similar vaccine responses to babies born to uninfected mothers by 9-12 months of age.	[123]
Helminth infections on Influenza vaccine	Mice	2022	*Litomosoides sigmodontis*	Influenza vaccine	Helminth-infected mice produced fewer antibodies in response to the flu vaccine, making it less effective. The presence of worms led to **↑** virus levels in the lungs despite vaccination, indicating **↓** vaccine efficacy.	[127]
*Trichinella spiralis* infection on RBD protein vaccine of SARS-CoV-2	Mice	2024	*Trichinella spiralis* infection and Albendazole treatment	mRNA COVID-19 vaccine	*Trichinella spiralis* impairs the efficacy of the COVID-19 vaccine. The infection suppresses immune responses to the vaccine, including **↓** levels of IgG, IgM, IgA, neutralizing antibodies, and splenic germinal center B cells. Treatment with albendazole (ALB) partially reverses this inhibitory effect, improving vaccine efficacy.	[120]
*S. mansoni* on measles vaccine	Human	2019	*S. mansoni* and praziquantel treatment	Measles vaccine	Infected children had **↓** anti-measles IgG levels and **↓** protection rates post-immunization; praziquantel treatment improved immune response.	[26]
*S. mansoni* on MVA85A TB vaccine in Ugandan adolescents	Human	2017	*S.mansoni*	MVA85A (TB vaccine)	TB vaccine immunogenicity was not affected by *S.mansoni* infection. No safety concerns, but *S.mansoni* infected had higher pre-vaccine IgG4.	[128]
Albendazole treatment and Influenza vaccine in Gabonese children	Human	2015	Albendazole	Seasonal influenza vaccine	Albendazole treatment had no significant impact on Influenza vaccine response; slight trend toward better immunogenicity in treated group.	[129]
Anthelmintic treatment during pregnancy and infant vaccine response	Human	2017	Albendazole, Praziquantel	DTP, HiB, hepatitis B vaccine	No significant effect of treatment on infant vaccine responses but strongyloidiasis was linked with enhanced vaccine response.	[130]
Helminths and multiple vaccine responses in Ugandan adolescents	Human	2024	*S. mansoni,* hookworm	BCG, yellow fever, typhoid, HPV, diphtheria vaccine	Schistosoma reduced responses to BCG and typhoid vaccines; hookworm increased diphtheria IgG but reduced HPV-16 IgG.	[131]
Helminths and GMZ2 malaria vaccine efficacy in malaria-endemic adults	Human	2021	*Schistosoma haematobium*, Strongyloides stercoralis	GMZ2 malaria vaccine	Schistosoma infection was linked to earlier malaria episodes; helminths reduced vaccine immunogenicity and efficacy.	[132]
Helminth exposure and response to Ebola virus glycoprotein antibody (Ad26.ZEBOV, MVA-BN-Filo vaccine regimen)	Human	2024	*Schistosoma mansoni*, *Acanthocheilonema viteae*, *Strongyloides ratti*	Ebola Virus Disease (EVD), response to Ad26.ZEBOV, MVA-BN-Filo vaccine regimen	No significant association between helminth exposure (via ELISA markers) and antibody concentration to EBOV GP post-vaccination. Five immune markers (CCL2/MCP1, FGFbasic, IL-7, IL-13, and CCL11/Eotaxin) were significantly lower in participants with helminth exposure but these did not correlate with vaccine response.	[133]

This table represents the key findings of some studies selected studies on the interaction between helminths and vaccine efficacy. It also includes the type of study, year of publication, the main findings, and references. ↑ stands for increase; ↓ stands for decrease.

## Data Availability

The data presented in this study are available in this article and Appendix A.

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
