# Peer review of "Helminth Coinfections Modulate Disease Dynamics and Vaccination Success in the Era of Emerging Infectious Diseases"

_vaccines, 2025, doi:10.3390/vaccines13050436_

Round 1

Reviewer 1 Report

Comments and Suggestions for Authors

This manuscript presents a comprehensive review of available literature relating to evidence of immunomodulatory effects of helminth infections on coinfections by various viruses and bacteria, as well as observed effects on vaccine efficacy. The authors provide details on what previous studies revealed relating to potential mechanisms of immune response alteration, and graciously organized observed differences according to type of coinfection (e.g., respiratory vs. blood-born viruses, type of pathogenic effects altered by helminth presence, and altered effects of vaccination). The authors further provide Tables detailing supporting evidence and observed effects with references, greatly facilitating further investigations. In addition, the summary conclusions and limitations of the present work are well presented, along with recommendations for uniformity in design of future research investigations. Overall, I highly recommend acceptance of this important review.

Author Response

We sincerely appreciate your thorough and insightful review of our manuscript. Your positive feedback on the organization, depth, and clarity of our review is highly appreciated. We are grateful for your recognition of our effort in systematically presenting the immunomodulatory effects of helminth infections on coinfections and vaccine efficacy, as well as in structuring the findings to facilitate further research.

Your acknowledgment of our detailed tables, summary conclusions, and recommendations for future research uniformity further validates our approach. We are delighted that you found our work valuable and worthy of recommendation for acceptance.

Reviewer 2 Report

Comments and Suggestions for Authors

This is a well-written review of the effects of helminth infection on bacterial and viral vaccine-induced immunity. The topic is relevant as more vaccines are developed for regions that are endemic for helminths. A major concept is to deworm the individuals or animals before vaccination because the worms tend to skew the responses to a Th2 bias and immune suppression. In this review, they point to a few studies where it was demonstrated that the immunosuppression remains for a longer period of time after deworming, which accounted for the vaccine failure. Helminth-infected pregnant women can transfer the immunosuppression to the fetus through placental transfer, and it concluded that vaccination in the pediatric population could be affected. The interpretation, summary, and conclusion raised in this review are noteworthy, needing additional research in this area.  

Author Response

We sincerely appreciate your thoughtful review and positive feedback on our manuscript. We are pleased that you found our review well-written and relevant, particularly in the context of vaccine development for helminth-endemic regions.

Your recognition of the key concepts discussed such as the impact of helminth-induced immune modulation, the potential long-term effects of immunosuppression post-deworming, and the maternal transfer of immunosuppressive effects reinforces the importance of further research in this field. We are especially grateful for your acknowledgment of the need for additional studies to better understand these mechanisms and improve vaccination strategies.

Thank you once again for your valuable insights and for highlighting the significance of our work.

Reviewer 3 Report

Comments and Suggestions for Authors

This article comprehensively summarizes the immune-modulatory effects of helminths during various diseases and vaccination. It is clearly written and easy to follow. However, there are a few important points to address.

I am concerned about the statistical analysis of the results. The authors present responses in percentages as beneficial or harmful. This can be misleading. As the authors rightly mentioned in multiple occasions that these studies are different, some in animal models, others in humans; there are large heterogeneities in study populations, helminths, viruses, and vaccines. Therefore, indicating as if helminths infection confers 86% protection against respiratory viral infections, is not only incorrect, but also misleading. What the authors are showing is, out of X number of articles investigating this, Y% of the article suggested immune modulation by helminths to have some beneficial effects during viral infection. Therefore, I strongly recommend removing all these percentage calculations and misleading data throughout the manuscript.

The authors need to keep restating whether the study was conducted in humans or in animal models while explaining them in the text, throughout the manuscript. Often, generalizing outcomes of diverse pathogens and different diseases as if universally approved cause-effects relationship is wrong. I recommend authors carefully revise the manuscript to avoid these statements. This diversity needs to be clearly stated in the limitations section as well.

This reviewer also would like to recommend using some alternative words rather than ‘protection’, used in many instances as if to protect against respiratory viral infections we need to set up helminth infection in humans. While there appears to be some beneficial effects of preexisting helminth infection, representing it as if ‘having helminth infection is good against respiratory viral infection’ is wrong because helminth infection has its own toll on human health. Rather, exploring those mechanisms to improve human health during various disease conditions should be the focus of the review.  

Overall, a careful revision to avoid too much generalization of the highly complex and context-specific interaction of helminth infection, various diseases, and vaccines is recommended. 

Author Response

We sincerely appreciate your insightful comments, and the time spent to provide detailed feedback on our manuscript. Your observations have been invaluable in refining our work and ensuring clarity in the interpretation of our findings.

While the synthesis of findings on viral and bacterial co-infections, as well as vaccine efficacy, introduces some heterogeneity in our review, we aimed to minimize this by organizing our discussions into thematic areas such as helminth-respiratory virus co-infection, helminth-bloodborne viral co-infection, helminth-bacterial co-infection, and helminth-vaccine interactions. This structure helped us prevent generalized conclusions and maintain clarity in our analysis.

We fully acknowledged that the diversity in study designs, ranging from observational studies and cross-sectional analyses to animal models, limits the robustness of direct comparisons as each approach has distinct limitations. This was highlighted in our review’s limitations section. We highlighted that observational and cross-sectional studies are informative but lack causative insights, while animal models may not fully replicate human immune responses to co-infections (see lines 699–703).

Regarding statistical misinterpretation, we have removed all percentage calculations that could be misleading. Instead, we have clarified that our synthesis reflects studies reporting specific immune-modulation and their effects on disease outcome rather than absolute protection conferred by helminth infections.

Additionally, we have thoroughly revised our manuscript to ensure that each study is clearly identified as either human or animal-based throughout the text (See lines 267,273,395,399,480,486,644). To avoid any implication that helminth infections are inherently beneficial, we have replaced the term "protection" with alternative terms such as "modulation," "impact," "less severe symptoms," or "severity mitigation." These changes are reflected throughout the manuscript, particularly in lines 255, 262, 271, 274, 352, 359, 502, 515, 540, and 559.

Furthermore, our review explored and discussed how the immunomodulatory effects of helminths could be leveraged to improve vaccine efficacy and guide vaccine development. We highlight that helminth-derived molecules such as ES-62 and glycans offer valuable templates for synthetic adjuvant design, as they interfere with innate immune pathways like NF-κB and TLR signaling. These mechanisms, while potentially dampening traditional vaccine responses, provide critical insights into developing novel adjuvants that can modulate immune responses in a more balanced manner. Previous studies have shown that rOv-ASP-1, a protein derived from the Onchocerca volvulus parasite, serves as an effective adjuvant, enhancing the efficacy of vaccines composed of proteins or synthetic peptides.

Mimicking helminth products like ES-62, which regulate inflammatory pathways, could enable the synthesis of vaccine molecules that enhance antigen-specific responses while simultaneously reducing adverse inflammation. This dual functionality is particularly advantageous in creating vaccines that are both effective and safer, with reduced side effects. Such approaches could include combining helminth-inspired adjuvants with immune stimulants, such as Toll-like receptor (TLR) agonists, to promote Th1 responses while the helminth-derived components mitigate excessive inflammation. Alternatively, derivatives of helminth molecules such as glutathione S-transferase from Fasciola hepatica (nFhGST) could be synthesized to retain their ability to modulate NF-κB or TLR pathways while avoiding complete suppression of immune activation. Structural modifications of helminth-derived molecules could also be employed to selectively retain their anti-inflammatory effects while reducing their capacity to suppress critical immune responses, such as Th1 or Th17 pathways. This strategy would ensure a balanced immune activation, preserving the ability to generate robust vaccine-induced immunity. The goal is to leverage the immunosuppressive capabilities of helminth-derived molecules in a selective manner minimizing harmful inflammation without compromising the immune responses necessary for protection and vaccine efficacy. Advanced techniques, including molecular docking, bioinformatics, and synthetic biology, can play a pivotal role in achieving this delicate balance, paving the way for innovative and efficient vaccine designs. These discussions and findings are detailed in lines 316–342.

Once again, we deeply appreciate your thoughtful feedback, which has strengthened our manuscript. Thank you for your valuable contribution.

Round 2

Reviewer 3 Report

Comments and Suggestions for Authors

All the comments raised earlier are addressed.